# Influences of Government Policies and Farmers' Cognition on Farmers' Participation Willingness and Behaviors in E-Commerce Interest Linkage Mechanisms during Farmer–Enterprise Games

**Xiaolu Wei and Junhu Ruan ***

College of Economics and Management, Northwest A&F University, Yangling 712100, China
* Correspondence: rjh@nwsuaf.edu.cn

**Abstract:** E-commerce interest linkage mechanisms serve as an effective solution to the problems of farmer–market cooperation, agricultural supply-side reforms, and farmers' income growth. This study, guided by the theory of planned behavior, undertook an evolutionary game analysis of farmer–enterprise cooperation with government interventions with farmers. Based on data from 554 questionnaires administered in Mei County, Shaanxi Province, China, this study found a difference between the realistic and optimal choices of farmers. In addition, this study used a structural equation model to investigate the influence of government policies and farmers' cognition on the participation willingness and behaviors of farmers in e-commerce interest-linkage mechanisms. The results showed that the optimal choice for farmers in a farmer–enterprise cooperative game is participation in e-commerce, and government policies can be used to improve farmer–enterprise e-commerce interest-linkage mechanisms. Farmers' basic characteristics and experiences impacted their cognition of e-commerce, which, in turn, had a significant positive effect on their e-commerce participation willingness and behaviors. Government policies had a positive effect on farmers' experiences, cognition of e-commerce, and participation behaviors, but no direct positive impact on farmers' willingness to participate. Government policies and farmers' basic characteristics interacted and acted together on the participation willingness and behavior of farmers.

**Keywords:** e-commerce interest linkage; participation willingness and behaviors; government policies; farmers' cognition; evolutionary game model; structural equation model

## 1. Introduction

The rapid growth of the economy and the fast rise of internet enterprises in China in recent years have led to tremendous changes in domestic business models. E-commerce is gradually developing in rural areas that are otherwise dominated by traditional sales methods. In 2005, the Chinese government proposed e-commerce business models, and in 2022, it provided clear, specific, and long-term development paths for rural e-commerce and digital rural areas: continue to promote the integrated development of primary, secondary, and tertiary industries in rural areas; encourage various regions to expand various agricultural functions; explore the diversified values of rural areas; and focus on the development of rural e-commerce. E-commerce is now accelerating its penetration into rural areas, with China's rural online retail sales reaching CNY 2.05 trillion in 2021, up 11.3 percent year on year. Various types of e-commerce interest linkage (EIL) mechanisms were established in the Yangtze Triangle area, Greater Bay area, and Chinese central and western areas, and rural e-commerce is booming. Rural e-commerce not only changes the models for the sale of agricultural products but also facilitates employment, income, and other aspects in rural areas. However, most farmers currently

continue to sell agricultural products through traditional methods [1]. Their willingness to participate and rural e-commerce behaviors are strongly impacted by their cognition of e-commerce and government policies. For this reason, research is needed on the factors that influence government policies and the impact of farmers' cognition on farmers' participation willingness and behaviors regarding EIL mechanisms [2,3]. This research, which used a farmer–enterprise cooperative game, can play a vital role in the implementation of agricultural supply-side reforms and the development of the rural economy.

Different from the agricultural food market electronic trading platforms [4] with industry chain-type, intermediary, professional, and alliance-type operation models in the United States and Europe, multiple types of interest linkage models have taken shape regarding lands, funds, labor, technologies, and sites as core elements in China; among these, the "farmer + enterprise" model is relatively common and effective [5].

Most studies on e-commerce participation explore factors that significantly influence the operation of these mechanisms from the perspective of enterprises [6]. The factors explored include the logistics service quality [7] and soundness of product supply chains, [8] development of scale economy and market internet coverage rate, [9] internal and external environment assessment of enterprises, logistics target setting, and strategic supply of supply chains [10]. In their research from the perspective of farmers, Cui et al. found that farmers' cognitive dimensions [11], social innovation [12], endowment, and regional environment [13] have a notable impact on their willingness to participate in e-commerce. Zhang found that joining a cooperative can decrease farmers' willingness to participate in e-commerce [14]. Luo et al. found that farmers' age, level of education [15], family income [16], personal characteristics [17], transaction costs [18], and other factors exert effects on farmers' participation behaviors regarding e-commerce. In terms of research methods, most scholars used a multivariate logistic model, structural equation model, and structural equation model, while a few scholars studied the related problems of supply chain coordination and put forward coordination schemes through evolutionary game theory [19–21].

As noted, most existing research on farmers' participation in e-commerce mechanisms takes the position of enterprises as primary. There is, by contrast, little game theory research on the participation behaviors of farmers and enterprises regarding EIL mechanisms. Although there is current research on the factors that influence their willingness to participate and their engagement in e-commerce, the perspectives are singular, and the farmers' willingness and behaviors are not combined for analysis. The multivariate logistic, interpretative structural, and structural equation models are the methods that are most used. Concrete reasons for variations in farmers' participation willingness and behaviors in e-commerce and reality have not been studied in any depth using the evolutionary game model.

Therefore, compared with the relevant literature, the novelty of this study lay in the use of the theory of planned behavior, which was applied to conduct an evolutionary game analysis on farmer–enterprise interest linkage mechanisms with government interventions and farmers as the subject. The game analysis results, along with the results of previous empirical studies of farmers in fruit planting (such as kiwi and strawberry) in Mei County, Shaanxi Province, revealed a difference between farmers' realistic and optimal choices. The structural equation model was then used to estimate and examine the causal relationship between farmers' characteristics (F1), experiences (F2), cognition (F3), behaviors (F4), willingness (F5), and government policies (F6). A corresponding analysis was carried out to put forward future directions.

## 2. Materials and Methods

### 2.1. Theoretical Hypotheses

Ajzen's theory of planned behavior predicts that all factors influencing behavior do so indirectly via behavioral intention. According to this theory, indicators such as behavioral attitude, external subjective norms, and perceived behavioral control can be used to measure the extent of behavioral intention to participate. For this reason, it was assumed in this study that with EIL mechanisms, farmers' willingness to participate and their e-commerce behaviors are influenced by internal and external factors, including farmers' basic characteristics and cognition, as well as government policies; various influencing factors were thus analyzed in this case.

Farmers' willingness to participate in EIL mechanisms is highly correlated with their behavior; this relationship was previously fully considered in theoretical and empirical analysis and will thus not be considered here. The theoretical game analysis concluded that farmers' optimal choices are inconsistent with reality. Hence, farmers' cognition, government policies, and other factors influencing farmers' participation willingness and behaviors in relation to EIL mechanisms were further considered.

### 2.2. Research Hypotheses

#### 2.2.1. Farmers' Characteristics

Farmer characteristics can be thought of as their intrinsic resources. Farmers with a greater resource endowment have a comparative advantage and are able to enhance their competitiveness in market transactions. Income, level of education, social relations, and other characteristics have a significant impact on farmers' participation in cooperative economic organizations [22–24]. On this basis, the research hypothesis H1 was proposed.

#### 2.2.2. Experiences

Experiences represent farmers' valuable knowledge about relevant technologies and markets accumulated in past agricultural production and operation activities they were occupied in, including transferring the external training into self-ability, long-term labor accumulation, and other activities. These previous experiences help farmers to know about e-commerce and accumulate knowledge, which then influences their participation willingness and behaviors regarding e-commerce [25,26]. On this basis, the research hypothesis H2 was proposed.

#### 2.2.3. Government Policies

Government policies are crucial external factors to guarantee farmers' participation in EIL mechanisms. Governments have established agricultural cooperatives and other village organizations [27,28], promoted farmer–enterprise cooperation [29,30], and other policies to improve the level of farmers' knowledge of e-commerce mechanisms, thereby facilitating their participation in them [17,31–33]. On this basis, the research hypotheses H5–H8 were proposed.

#### 2.2.4. Farmers' Cognition

Farmers' cognition concerns farmers' self-perception of the level of difficulty, marketing effects, and development anticipation of participating in EIL mechanisms. As more and more farmers start to learn about online shopping, their attitude toward the Internet and information is gradually changing [34]; problems of distrust, security concerns, and their sense of loyalty have improved; and their cognitive attitudes toward participation are more positive, making them more likely to take part in e-commerce mechanisms [35–38]. On this basis, the research hypotheses H3 and H4 were proposed.

2.2.5. Interaction

Under the theory of planned behavior, three variables, namely, behavioral attitudes, external subjective norms, and perceived behavioral control, interact and act together on farmers' participation willingness and behaviors regarding e-commerce. On this basis, the research hypothesis H9 was proposed. Based on the abovementioned hypotheses, several research hypotheses were formulated and are shown in Table 1.

**Table 1.** Research hypotheses.

| Hypothesis Path | Hypothesis Content | Supporting References |
|---|---|---|
| H1 | Farmers' basic characteristics positively influence their cognition of e-commerce. | [22–24] |
| H2 | Farmers' experiences positively influence their cognition of e-commerce. | [25,26] |
| H3 | Farmers' cognition of e-commerce positively impacts their participation willingness. | [34–38] |
| H4 | Farmers' cognition of e-commerce positively impacts their participation behaviors regarding e-commerce. | [34–38] |
| H5 | Government policies positively affect farmers' experiences. | [17,31–33] |
| H6 | Government policies positively affect farmers' cognition of e-commerce. | [17,27–33] |
| H7 | Government policies exert a positive influence on farmers' participation willingness. | [39,26] |
| H8 | Government policies exert a positive influence on farmers' participation behaviors regarding e-commerce. | [17,31–33] |
| H9 | Government policies interact with farmers' basic characteristics and together act on farmers' participation willingness and behaviors regarding e-commerce. | The theory of planned behavior |

*2.3. Theoretical Models*

In line with the aforesaid theoretical and research hypotheses, a model of farmers' participation willingness and behaviors regarding EIL mechanisms was constructed and is depicted in Figure 1.

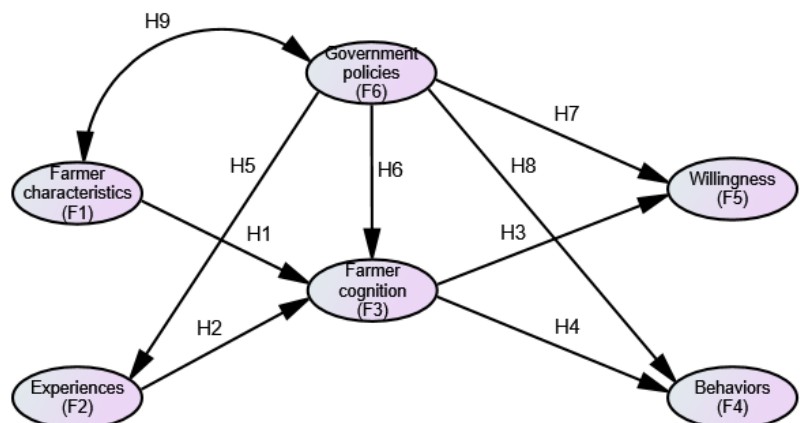

**Figure 1.** Model of farmers' participation willingness and behaviors regarding EIL mechanisms.

*2.4. Variable Descriptions*

Shaanxi Province, a major province of traditional agriculture in China, actively responds to the national call to encourage the construction and development of digital agricultural and rural systems and advances the establishment of rural EIL mechanisms. This research group planned to analyze influencing factors in the regions where e-commerce systems had sound development but were less adopted by farmers and put

forward targeted suggestions to provide experience for the development in other regions. Therefore, empirical research was carried out on 56 villages in one street and seven towns in Mei County, Baoji City, Shaanxi Province, China, from March to May 2022, employing questionnaires, interviews, and other survey sampling methods. A total of 604 questionnaires were distributed, and 554 valid questionnaires were recovered, with an effectiveness rate of 91.7%. The proportion of agricultural products examined by the sample group that were subject to e-commerce was 66%.

IBM SPSS 26.0 was employed to conduct the descriptive analysis (Tables 2 and 3) using six variables and 19 items in the model of farmers' participation willingness and behaviors in EIL mechanisms. As Table 2 shows, the indicator of participation behaviors took values of 0 or 1, while the remaining items had more than three options on average.

**Table 2.** Sample basic information statistics.

| Variable Name | Classification | Sample Number | Frequency (%) |
|---|---|---|---|
| Gender | Male | 389 | 70.1 |
| | Female | 166 | 29.9 |
| Age | Under 30 years old | 216 | 38.9 |
| | 30–40 years old | 238 | 42.9 |
| | 40–50 years old | 68 | 12.3 |
| | 50–60 years old | 29 | 5.2 |
| | Over 60 years old | 4 | 0.7 |
| Level of education | Zero | 5 | 0.9 |
| | Elementary school | 17 | 3.1 |
| | Junior high school | 124 | 22.3 |
| | Senior high school | 224 | 40.4 |
| | College or above | 185 | 33.3 |
| Planting year | Within 3 years | 29 | 5.2 |
| | 3–5 years | 43 | 7.7 |
| | 5–10 years | 256 | 46.1 |
| | 10–20 years | 106 | 19.1 |
| | Over 20 years | 121 | 21.8 |

**Table 3.** Variable definitions and descriptive statistics.

| Variable Name | Item | Variable Definition and Assignment | Average | Standard Deviation |
|---|---|---|---|---|
| Participation behaviors | Participation in EIL mechanisms or not | 1 = participation, 0 = non-participation | 0.66 | 0.475 |
| Willingness to participate | Willingness to participate in EIL mechanisms | 1 = absolutely not, 2 = partially not, 3 = normally, 4 = partially willing, 5 = totally willing | 3.70 | 1.119 |
| Willingness to encourage surrounding people to participate | Willingness to encourage acquaintances to participate in EIL mechanisms | 1 = absolutely not, 2 = partially not, 3 = normally, 4 = partially willing, 5 = totally willing | 3.66 | 1.093 |
| Daily online time | Daily online time | 1 = 0–2 h, 2 = 2–4 h, 3 = 4–6 h, 4 = 6–8 h, 5 = over 8 h | 4.18 | 0.818 |
| Social relations | Association with relatives and friends who participate in e-commerce | 1 = never, 2 = sometimes, 3 = normally, 4 = often, 5 = always | 4.00 | 0.926 |
| Level of education | Level of education | 1 = zero, 2 = elementary school, 3 = junior high school, 4 = senior high school, 5 = college or above | 4.02 | 0.873 |
| Age | Age interval | 1 = over 60 years old, 2 = 50–60 years old, 3 = 40–50 years old, 4 = 30–40 years old, 5 = | 4.14 | 0.876 |

| | | under 30 years old | | |
|---|---|---|---|---|
| Planting year | Agricultural planting year | 1 = within 3 years, 2 = 3–5 years, 3 = 5–10 years, 4 = 10–20 years, 5 = over 20 years | 3.45 | 1.074 |
| E-commerce training | Frequency of participation in e-commerce training | 1 = never, 2 = sometimes, 3 = normally, 4 = often, 5 = always | 3.46 | 1.173 |
| Technical training | Frequency of participation in planting technical training | 1 = never, 2 = sometimes, 3 = normally, 4 = often, 5 = always | 3.42 | 1.189 |
| Cognition of EIL mechanisms | Cognition of EIL mechanisms | 1 = absolutely not, 2 = partially not, 3 = normally, 4 = partially, 5 = totally | 3.32 | 1.194 |
| Mastery of and proficiency in e-commerce operation | Mastery of and proficiency in e-commerce operation | 1 = absolutely not, 2 = partially not, 3 = normally, 4 = partially, 5 = totally | 3.63 | 1.240 |
| Cognition of marketing effects of e-commerce platforms | Marketing effects of e-commerce platforms | 1 = very poor, 2 = partially poor, 3 = normal, 4 = partially good, 5 = very good | 3.49 | 1.123 |
| E-commerce future development anticipation | E-commerce future development anticipation | 1 = very poor, 2 = partially poor, 3 = normally, 4 = partially good, 5 = very good | 4.06 | 0.836 |
| Understanding of government e-commerce policies | Understanding of government policies on e-commerce | 1 = absolutely not, 2 = partially not, 3 = normally, 4 = partially, 5 = totally | 3.58 | 1.183 |
| Publicity | Government publicity for policies on e-commerce | 1 = very small, 2 = partially small, 3 = normally, 4 = partially large, 5 = very large | 3.38 | 1.217 |
| Subsidies | Government subsidy for e-commerce participation | 1 = never heard of nor accepted, 2 = heard of but not accepted, 3 = heard of and partially accepted, 4 = heard of and totally accepted | 3.08 | 1.348 |
| Training | E-commerce and technical training organized by the government | 1 = never heard of nor accepted, 2 = heard of but not accepted, 3 = heard of and partially accepted, 4 = heard of and totally accepted | 3.08 | 1.334 |
| Supervision | Government supervision of e-commerce policies | 1 = very small, 2 = partially small, 3 = normally, 4 = partially large, 5 = very large | 3.24 | 1.229 |

*2.5. Model Construction*

2.5.1. Evolutionary Game Model

Based on bounded rationality, evolutionary game theory goes against the assumption of the perfect rationality of economic actors in traditional game theory. The choice of strategy of game subjects is continuously adapting and tends to be locally stable in the end. In this study, game subjects are farmers and enterprises and, according to evolutionary game theory, they act under bounded rationality. In addition, government interventions are included in the game, and farmers who participate in EIL mechanisms are subject to only positive external influences, such as technologies and publicity. The strategy space of farmers is {participation in mechanisms, non-participation in mechanisms}, while that of enterprises is {providing platforms, not providing platforms}. Their strategy selection is shown in Table 4.

**Table 4.** Selectable strategies of farmers and enterprises.

| Participant | Selectable Strategy | |
|---|---|---|
| Farmers | Participation | Non-participation |
| Enterprises | Providing platforms | Not providing platforms |

The probability of farmers' participation in EIL mechanisms is expressed as $x$ and $(1 - x)$ refers to the probability of their non-participation. The probability of enterprises providing platforms is denoted by $y$, and $(1 - y)$ denotes the probability of not providing platforms; specifically, $(0 \leq x, y \leq 1)$. $R_1$ signifies the available revenue of farmers in the case of participation in EIL mechanisms, $R_2$ signifies the available reve-

nue of farmers in the case of non-participation in EIL mechanisms, $R_3$ signifies the available revenue of enterprises in the case of providing platforms, and $R_4$ signifies the available revenue of enterprises in the case of not providing platforms. $C_1$ represents the costs of farmers in the case of participation in EIL mechanisms, $C_2$ represents the costs of farmers in the case of non-participation in EIL mechanisms, $C_3$ represents the costs of enterprises in the case of providing platforms, and $C_4$ represents the costs of enterprises in the case of not providing platforms. S represents positive government influences on farmers' participation in EIL mechanisms and P is the resource wasting caused by enterprises' providing platforms and farmers' not participating.

2.5.2. Structural Equation Model

In this study, the structural equation model was used. This overcomes the limitations of the general linear regression method and is applicable to the research of multiple variables with joint action. A total of six variables, namely, farmers' characteristics (F1), experiences (F2), cognition (F3), behaviors (F4), willingness (F5), and government policies (F6), were assumed as latent variables. In addition, 19 items in the questionnaires were selected as observed variables reflecting these latent variables. Path analysis was conducted by constructing the structural equation model to test the causal relationships between the latent variables. The general form of the model is stated as Equation (1):

$$Y = \alpha Y + \beta X + \varepsilon \tag{1}$$

where $Y$ expresses the endogenous variable vectors, $X$ expresses the exogenous variable vectors, $\alpha$ is a structural coefficient matrix that represents the relationships between endogenous variables, $\beta$ is a structural coefficient matrix that represents the influences of exogenous variables on endogenous variables, and $\varepsilon$ is a residual term that represents the unexplained parts.

**3. Results and Discussion**

*3.1. Evolutionary Game Model*

First, the evolutionary game model of the farmer–enterprise interest linkage mechanisms under government interventions was analyzed.

3.1.1. Model Analysis

Following the method proposed by Friedman, a local stability analysis was undertaken using the interest payoff matrix (shown in Table 5) and the Jacobian matrix to explore the evolutionarily stable strategy (ESS) formed by both sides of the games.

**Table 5.** Game payoff matrix for the strategy selection of farmers and enterprises with government intervention.

| Farmer | Enterprise | |
|---|---|---|
| | Providing platforms | Not providing platforms |
| Participation in the linkage mechanisms | $R_1 - C_1 + S, R_3 - C_3$ | $R_2 + S, R_4 - C_4$ |
| Non-participation in the linkage mechanisms | $R_2 - C_2, R_3 - C_3 - P$ | $R_2 - C_2, R_4 - C_4$ |

Following the correlation theory of evolutionary games, the calculation properties of expected revenues, the expected revenues of farmers in the case of participation $U_1$ and in the case of non-participation $U_2$ could be expressed as follows:

$$U_1 = y(R_1 - C_1 + S) + (1 - y)(R_2 + S) = y(R_1 - C_1 - R_2) + R_2 + S \tag{2}$$

$$U_2 = y(R_2 - C_2) + (1 - y)(R_2 - C_2) = R_2 - C_2 \tag{3}$$

The average expected revenue of the farmers was

$$U_A = xU_1 + (1-x)U_2 \tag{4}$$

Based on the Malthusian dynamic equation, the replicated dynamic equation ($t$ represents time) of the probability $x$ of farmers choosing the "cooperation" strategy was

$$F(x) = \frac{dx}{dt} = x(U_1 - U_A) = x(1-x)[y(R_1 - C_1 - R_2) + S + C_2] \tag{5}$$

Similarly, the expected revenue of enterprises in the case of providing platforms $U_3$ and not providing platforms $U_4$ was expressed as follows:

$$U_3 = x(R_3 - C_3) + (1-x)(R_3 - C_3 - P) = xP + R_3 - C_3 \tag{6}$$

$$U_4 = x(R_4 - C_4) + (1-x)(R_4 - C_4) = R_4 - C_4 \tag{7}$$

The average expected revenues of the enterprises was

$$U_B = yU_3 + (1-y)U_4 \tag{8}$$

The replicated dynamic equation of the probability $y$ of enterprises choosing the "cooperation" strategy was expressed as follows:

$$F(y) = \frac{dy}{dt} = y(U_3 - U_B) = y(1-y)(xP + R_3 - C_3 - R_4 + C_4) \tag{9}$$

The replicated dynamic equation group consisting of Equations (5) and (9) was expressed as follows:

$$\begin{cases} F(x) = \dfrac{dx}{dt} = x(U_1 - U_A) = x(1-x)[y(R_1 - C_1 - R_2) + S + C_2] \\ F(y) = \dfrac{dy}{dt} = y(U_3 - U_B) = y(1-y)(xP + R_3 - C_3 - R_4 + C_4) \end{cases}$$

With the replicated dynamic equation group set as $F(x) = \frac{dx}{dt} = 0$ and $F(y) = \frac{dy}{dt} = 0$, the following five local equilibrium points were obtained: $A(0,0)$, $B(0,1)$, $C(1,0)$, $D(1,1)$, and $E(x^*, y^*)$.

$$\left(x^* = \frac{R_4 - C_4 + C_3 - R_3}{P},\ y^* = \frac{S + C_2}{R_2 + C_1 - R_1}, \text{and } (0 \le x, y \le 1)\right)$$

$$\left(x^* = \frac{R_4 - C_4 + C_3 - R_3}{P},\ y^* = \frac{S + C_2}{R_2 + C_1 - R_1}, \text{and } (0 \le x, y \le 1)\right)$$

The Jacobian matrix obtained via the replicated dynamic equation group (Equations (5) and (9)) was expressed as

$$J = \begin{bmatrix} (1-2x)[y(R_1 - C_1 - R_2) + S + C_2] & x(1-x)(R_1 - C_1 - R_2) \\ y(1-y)P & (1-2y)(xP + R_3 - C_3 - R_4 + C_4) \end{bmatrix} \tag{10}$$

$$DetJ = (1-2x)[y(R_1 - C_1 - R_2) + S + C_2] \times (1-2y)(xP + R_3 - C_3 - R_4 + C_4)$$
$$-x(1-x)(R_1 - C_1 - R_2) \times y(1-y)P \tag{11}$$

$$TrJ = (1-2x)[y(R_1 - C_1 - R_2) + S + C_2] + (1-2y)(xP + R_3 - C_3 - R_4 + C_4) \tag{12}$$

With $DetJ > 0$ and $TrJ < 0$, the equilibrium point of the replicated dynamic equations was an ESS, which was obtained by considering the symbols of the determinants and the trace values of the Jacobian matrix at five local equilibrium points based on the assumed conditions and $(0 \le x, y \le 1)$.

According to the analysis results set out in Table 6, among the five local equilibrium points, only the point $D(1,1)$ was an ESS, indicating that the cooperation strategy was chosen by both farmers and enterprises. There were also three unstable points—$A(0,0)$,

$B(0,1)$, and $C(1,0)$—indicating that non-cooperation or other strategies were chosen by farmers and enterprises. Additionally, there was a saddle point at $E(x^*, y^*)$.

**Table 6.** Local stability analysis of the evolutionary game system with government intervention.

| Equilibrium Point | DetJ | Symbol | TrJ | Symbol | Stability |
|---|---|---|---|---|---|
| (0,0) | $(S + C_2)(R_3 - C_3 - R_4 + C_4)$ | − | $(S + C_2) + (R_3 - C_3 - R_4 + C_4)$ | +/− | Unstable |
| (0,1) | $(R_1 + S + C_2 - C_1 - R_2)(R_4 + C_3 - C_4 - R_3)$ | + | $(R_1 + S + C_2 - C_1 - R_2) + (R_4 + C_3 - C_4 - R_3)$ | +/− | Unstable |
| (1,0) | $-(S + C_2)(P + R_3 - C_3 - R_4 + C_4)$ | − | $-(S + C_2) + (P + R_3 - C_3 - R_4 + C_4)$ | +/− | Unstable |
| (1,1) | $(R_1 + S + C_2 - C_1 - R_2)(P + R_3 - C_3 - R_4 + C_4)$ | + | $-(R_1 + S + C_2 - C_1 - R_2) - (P + R_3 - C_3 - R_4 + C_4)$ | − | ESS |
| $(x^*, y^*)$ | E | + | 0 | | Saddle point |

Note: $E = \dfrac{(C_3 + R_4 - R_3 - C_4)(S + C_2)(P - C_3 - R_4 + R_3 + C_4)(R_2 + C_1 - R_1 - S - C_2)}{P(R_2 + C_1 - R_1)}$.

### 3.1.2. Evolutionary simulation

The dynamic evolutionary simulation of the game was carried out using MATLAB to allow for a visual assessment of the game between farmers and enterprises. The simulation cycles were set to 10, along with the following variables: $R_1 = 0.5$, $R_2 = 0.3$, $R_3 = 6$, $R_4 = 5$, $C_1 = 0.1$, $C_2 = 0.1$, $C_3 = 1$, $C_4 = 0.5$, $S = 0.5$, and $P = 0.1$. The initial values $(x, y)$ of the numerical simulation were set as $(0.3, 0.5)$, $(0.5, 0.5)$, and $(0.5, 0.8)$. The dynamic evolution process of the strategy selection of farmers and enterprises changing over time is displayed in Figure 2.

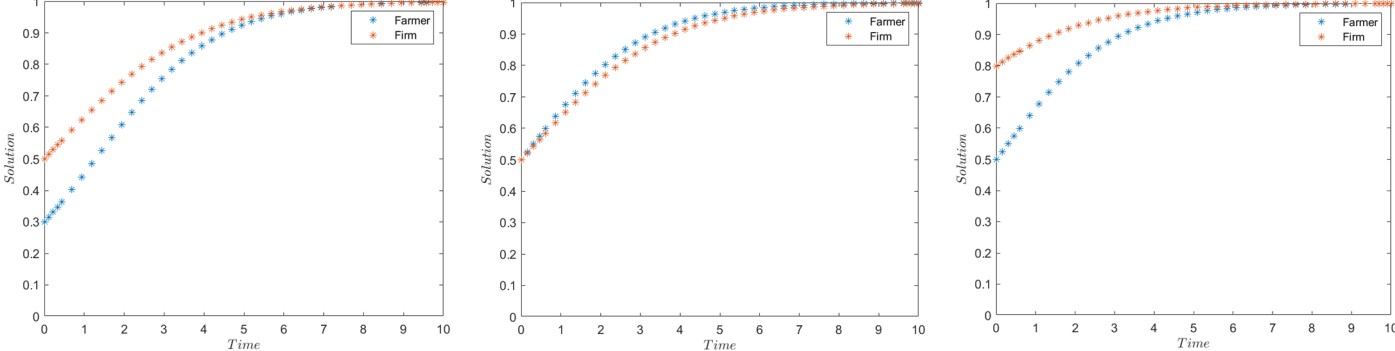

**Figure 2.** Dynamic evolution diagram of the strategy selection of farmers and enterprises.

As indicated in Figure 2, when the initial probability value was $(0.3, 0.5)$, that is, when the probability of farmer participation was less than the probability of enterprise participation, the final dynamic evolution strategy was both parties participating. When the initial probability value was $(0.5, 0.5)$, that is, when the probability of farmer participation was equal to the probability of enterprise participation, the final dynamic evolution strategy was both parties participating. When the initial probability value was $(0.5, 0.8)$, that is, when the probability of farmer participation was greater than the probability of enterprise participation, the final dynamic evolution strategy was both parties participating. Thus, the various initial probability values for the strategy selection of farmers and enterprises, i.e., $(x, y)$, resulted in final game evolution results that converged to the point $D(1,1)$, i.e., the ESS, meaning that the cooperation strategy was chosen by both farmers and enterprises.

From the government's point of view, the final evolution results for both sides of the games were also influenced by differences in the equation parameter S, as shown in Figure 3. With the other conditions unchanged and a change in the positive government in-

fluence S (set as 0.5, 1, and 2), regarding the farmers' participation in the linkage mechanisms, the evolutionary strategy of game subjects exhibited the trend depicted in Figure 3 with the adjustment of S. As depicted in Figure 3, as S increased, the change in the probability of farmers tending to cooperation became increasingly rapid. This indicated that the positive government influence S had a positive impact on the farmers' participation in the linkage mechanisms.

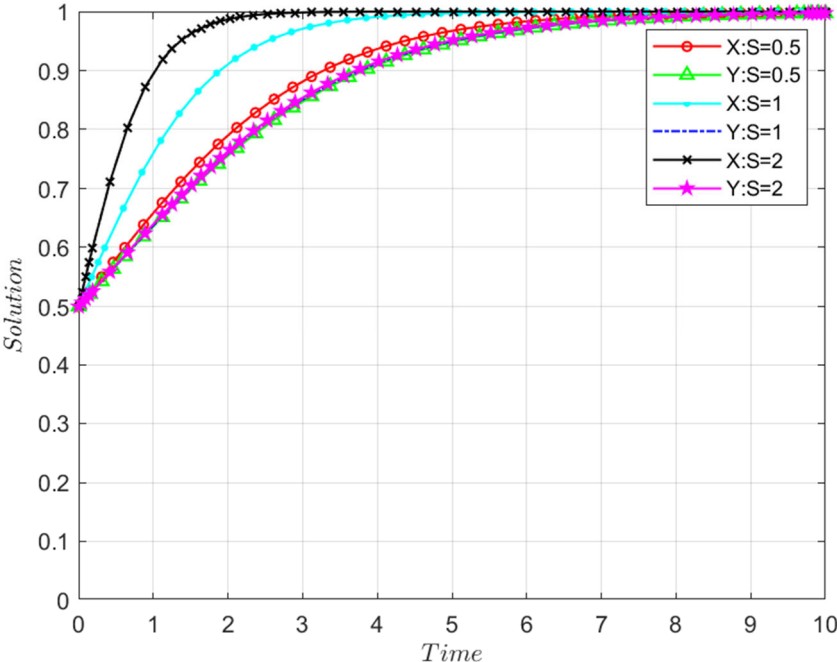

**Figure 3.** Trend of the farmer–enterprise evolutionary game strategy under government interventions.

In summary, the replicated dynamic equations $F(x) = \frac{dx}{dt} = 0$ and $F(y) = \frac{dy}{dt} = 0$ were solved through the construction of the evolutionary game model to obtain equilibrium points, and a local stability analysis was conducted to identify the ESS. Furthermore, MATLAB software was used for the numerical simulation. It was found that the optimal choice for farmers was to participate in the EIL mechanisms, and the implementation of government policies and systems could promote the improvement process of the EIL mechanisms for farmers and enterprises. This was contradictory to the reality that farmers' participation in the EIL mechanisms was not high. For this reason, a structural equation was constructed in terms of the farmers' basic characteristics and cognition and government policies to analyze the factors that affected farmers' participation willingness and behaviors.

### 3.2. Structural Equation Analysis

#### 3.2.1. Reliability and Validity Tests

Reliability and validity analysis of the data collected via the questionnaires were conducted using the software SPSS 26.0 and Amos 24.0, with the participation behaviors (PB) in e-commerce not involved since they had only one observed variable. The test results of the other five variables are displayed in Table 7. The Cronbach's alphas of behavioral willingness, farmers' characteristics, experiences, farmers' cognition, and government policies were greater than or close to 0.8, with good measurement reliability. As for the validity test, the load capacities of the standard factors, as well as the KMO values, were all greater than 0.6, and the convergent validity AVE was greater than or close to 0.5, revealing good validity.

**Table 7.** Reliability and validity analysis.

| Variable | Variable Setting | Dimensionality | Reliability | Validity | | |
| | | | Cronbach's $\alpha$ | Load Capacity of Factors | KMO | AVE |
|---|---|---|---|---|---|---|
| F1 | FCS1 | Daily online time | | 0.766 | | |
| | FCS2 | Social relations | 0.893 | 0.932 | 0.794 | 0.689 |
| | FCS3 | Level of education | | 0.94 | | |
| | FCS4 | Age | | 0.645 | | |
| F2 | EXP1 | Planting year | | 0.614 | | |
| | EXP2 | E-commerce training | 0.798 | 0.88 | 0.662 | 0.589 |
| | EXP3 | Technical training | | 0.785 | | |
| F3 | COG1 | Cognition of EIL mechanisms | | 0.619 | | |
| | COG2 | Mastery of and proficiency in e-commerce operation | | 0.83 | | |
| | COG3 | Cognition of marketing effects of e-commerce platforms | 0.838 | 0.658 | 0.825 | 0.528 |
| | COG4 | E-commerce development anticipation | | 0.87 | | |
| | COG5 | Understanding of government policies on e-commerce | | 0.614 | | |
| F5 | WTP1 | Willingness to participate | | 0.813 | | |
| | WTP2 | Willingness to encourage acquaintances to participate | 0.856 | 0.921 | 0.600 | 0.755 |
| F6 | GP1 | Publicity | | 0.693 | | |
| | GP2 | Subsidy | | 0.617 | | |
| | GP3 | Training | 0.784 | 0.808 | 0.771 | 0.477 |
| | GP4 | Supervision | | 0.628 | | |

### 3.2.2. Model Fitting

The model fitness was judged using Amos 24.0. It was found that $\chi^2/df = 2.616$, meeting the standard value (smaller than 3), while $GFI = 0.935$ and $AGFI = 0.913$, both of which were greater than the ideal value (0.9). In addition, $RMSEA = 0.054$, which was smaller than 0.8 and met the ideal standard value, thus indicating an ideal absolute fit index. Furthermore, $CFI = 0.959$ and $TLI = 0.951$, both of which were greater than the ideal standard value (0.9), thus indicating an ideal value-added fit index. Therefore, the model had an ideal overall fitness, and the structural equation model was effective. The diagram of the final standardized path coefficients of the model is depicted in Figure 4.

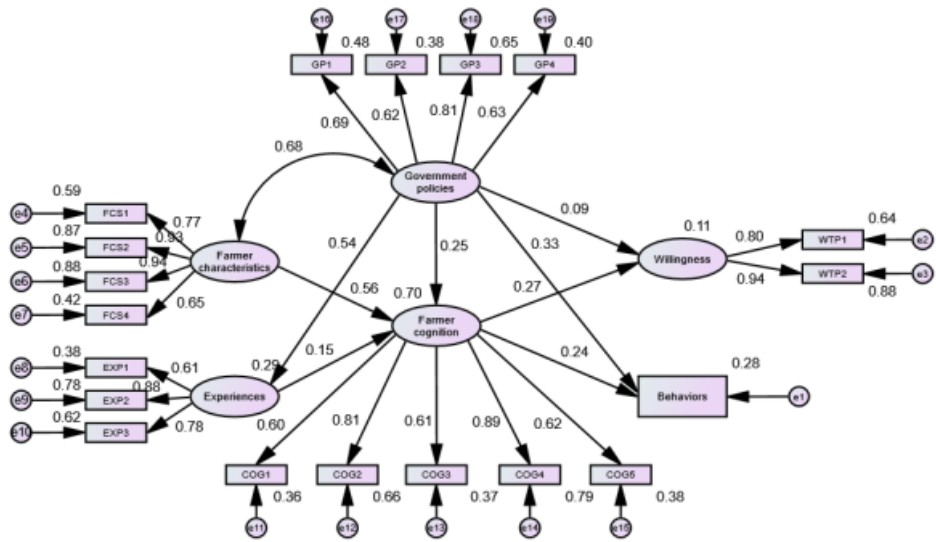

Chi-square=368.904 DF=141 Chi/DF=2.616 GFI=.935 AGFI=.913 RMSEA=.054

**Figure 4.** Diagram of the standardized path coefficients of the model.

### 3.2.3. Hypothesis Tests

According to Table 8, the path coefficient of government policies --> farmers' participation willingness in e-commerce (H7) failed to pass the significance test, with the absolute value of the statistic C.R. also being smaller than 2, suggesting that the relationship between the variables in the proposed H7 had no significant influence. The path coefficients of the other variables all passed the significance test, revealing that the relationships between the corresponding latent variables are significant.

**Table 8.** Path coefficients and their significance tests.

| Path | Non-Standardization | S.E. | C.R. | P | Standard Path Coefficient | Corresponding Hypothesis |
|---|---|---|---|---|---|---|
| F1 --> F3 | 0.720 | 0.079 | 9.063 | *** | 0.559 | H1 |
| F2 --> F3 | 0.121 | 0.033 | 3.688 | *** | 0.155 | H2 |
| F3 --> F5 | 0.325 | 0.094 | 3.437 | *** | 0.266 | H3 |
| F3 --> F4 | 0.156 | 0.043 | 3.642 | *** | 0.241 | H4 |
| F6 --> F2 | 0.598 | 0.064 | 9.336 | *** | 0.542 | H5 |
| F6 --> F3 | 0.220 | 0.051 | 4.354 | *** | 0.254 | H6 |
| F6 --> F5 | 0.094 | 0.080 | 1.178 | 0.239 | 0.089 | H7 |
| F6 --> F4 | 0.185 | 0.039 | 4.719 | *** | 0.329 | H8 |
| F1 <--> F6 | 0.325 | 0.034 | 9.500 | *** | 0.679 | H9 |

Note: F1—farmers' characteristics, F2—experiences, F3—cognition, F4—behaviors, F5—willingness, F6—government policies. The *** indicates that the statistical test has reached a 1% significance level.

### 3.2.4. Results Analysis

According to the analysis of the model results:

(1) Hypotheses H1 and H2 held, which was in agreement with the aforementioned hypothesis [22–26]. In other words, farmers' basic characteristics and experiences had a significant positive effect on their e-commerce cognition, with path coefficients of 0.559 and 0.155, respectively. This implied that farmers' characteristics had

a significant effect on the improvement of their cognition. The standardized path coefficients of the four latent farmer characteristic variables FCS1–4 were 0.766, 0.932, 0.940, and 0.645, respectively, showing that the farmers' level of education had the greatest positive effect, followed by social relations, daily hours online, and age. The standardized path coefficients of the latent variables of experiences EXP1–3 were 0.614, 0.880, and 0.785, respectively, indicating that e-commerce training exhibited the greatest positive effects on farmers' experiences, followed by technical training and planting year.

(2) Hypotheses H3 and H4 held, which was in agreement with the aforementioned hypothesis [34–38]. That is, farmers' e-commerce cognition positively impacted their participation willingness and behaviors in e-commerce, with path coefficients of 0.266 and 0.241, respectively, implying that farmers' participation willingness and behaviors in e-commerce were greatly affected by their cognition, which had a greater influence on their participation willingness. The standardized path coefficients of the latent variables of farmers' e-commerce cognition COG1–5 were 0.619, 0.830, 0.658, 0.870, and 0.614, respectively, signifying that e-commerce future development anticipation had the greatest positive effect on farmers' e-commerce cognition, followed by mastery of and proficiency in e-commerce operation, cognition of marketing effects of e-commerce platforms, cognition of EIL mechanisms, and understanding of government policies on e-commerce.

(3) Hypotheses H5, H6, and H8 held, which was in agreement with the aforementioned hypothesis [17,27–33]. Government policies positively affected farmers' experiences, cognition of e-commerce, and participation behaviors, with path coefficients of 0.542, 0.254, and 0.329, respectively, implying that farmers' e-commerce cognition and participation behaviors were greatly affected by the policies with the greatest influence on their experiences. However, hypothesis H7 did not hold, meaning that government policies had no direct positive influence on farmers' participation willingness, which was contrary to the extant literature [39,26]. They could, to some extent, indirectly affect farmers' participation willingness by affecting their basic characteristics, experiences, and cognition due to the lagging and weakening of the transmission mechanisms. As a result, farmers' participation willingness and behaviors in real life deviated from reality. The standardized path coefficients of the latent government-policy variables were 0.693, 0.617, 0.808, and 0.628, respectively, implying that government training had the greatest positive effect on government policies, followed by publicity, supervision, and subsidy.

(4) Hypothesis H9 held, which fits with the theory of planned behavior. In other words, government policies interacted with farmers' basic characteristics with a path coefficient of 0.679, implying that farmers' participation willingness and behaviors in e-commerce were jointly affected by government policies and their basic characteristics, in line with the theory of planned behavior.

### 3.3. Structural Equation Analysis

With the rapid development of the Internet, the cross-border integration of traditional agriculture and the Internet is inevitable. Therefore, in the EIL mechanisms, it is very important to determine the strategic choices of all parties. Based on the findings, the following conclusions were reached.

First, farmers' participation in the EIL mechanisms was the optimal choice in the farmer–enterprise cooperation games with government interventions, and the implementation of relevant government policies and systems could promote the improvement of the EIL mechanisms for farmers and enterprises.

Second, farmers' cognition of e-commerce was affected by their basic characteristics and experiences, and farmers' basic characteristics positively influenced the improvement of farmers' cognition in a more significant manner.

Third, farmers' cognition of e-commerce had a positive influence on their participation willingness and behaviors in e-commerce and exhibited a greater influence on their participation willingness.

Fourth, government policies had a significant positive effect on farmers' experiences, cognition of e-commerce, and participation behaviors, without any direct positive effect on their participation willingness.

Fifth, government policies and farmers' basic characteristics interacted and jointly acted on farmers' participation willingness and behaviors in e-commerce.

To some extent, this study made a more comprehensive demonstration of the previous relevant studies and combined the willingness and behavior from the perspective of farmers. A combination of an evolutionary game and structural equation model was used. Moreover, hypothesis H7 contradicted the existing literature [39,26] by showing that government policies had no direct impact on farmers' willingness to participate. However, at the same time, government policies can indirectly influence farmers' participation willingness in e-commerce by affecting their basic characteristics, experiences, and cognition. This explained the difference between farmers' realistic and optimal choices and showed that government policies were an external influencing factor of great importance.

## 4. Conclusions

Based on the theory of planned behavior, this study employed an evolutionary game analysis of farmer–enterprise cooperation with government intervention from the perspective of farmers. There was a difference between the realistic and optimal choices of farmers according to the evolutionary game analysis results combined with empirical research results. In addition, structural equation modeling was used to explore the impacts of government policies and farmers' cognition on farmers' participation willingness and behaviors regarding the EIL mechanisms. The following suggestions are proposed to increase farmers' participation willingness and behaviors regarding the EIL mechanisms.

Considering the great importance of government policies in encouraging farmers to participate in the EIL mechanisms, the relevant policies should be optimized, adjusted, and implemented. The government should attach great importance to e-commerce and technical training for farmers, increase publicity concerning e-commerce policies, grant further subsidies to e-commerce enterprises and participating farmers, promote the e-commerce knowledge and literacy of farmers, and improve farmers' satisfaction with e-commerce. In this way, farmers' participation willingness will be boosted, and the deviation between their participation willingness and behaviors will be reduced. Additionally, government departments should also strengthen everyday communication and interaction with farmers, understand the actual situation, and regulate and supervise with a view to securing effective protection of the interests of farmers.

Moreover, the government should constantly improve the construction of e-commerce infrastructure in rural areas and help to reduce the participation cost faced by farmers in e-commerce. According to the results here, farmers' cognition of e-commerce had a significant positive influence on their participation willingness and behaviors regarding e-commerce. The results revealed that farmers' self-perception of the level of difficulty, marketing effects, and development anticipation of participating in the EIL mechanisms were important factors that restricted the farmers' participation willingness and behaviors. Therefore, a diversity of policies should be adopted by governments at all levels in keeping with the local context to strengthen financial support for rural e-commerce; improve network infrastructures in rural areas, local logistics network design, and local logistics resource allocation; and reduce the cost to farmers of participation in e-commerce. In this way, the real interests of farmers will be ensured, and farmers will recognize that participation in e-commerce can increase their income. As a result, they will develop a positive attitude toward participating in e-commerce and



grow a considerable interest in the EIL mechanisms, thus facilitating their participation in e-commerce mechanisms.

All forces should be fully mobilized to allow for participation in the EIL mechanisms to boost the high-quality development of e-commerce in rural areas. It requires the radiation effects of big farmers and e-commerce enterprises and the leadership example of village officials for participation in e-commerce mechanisms. This will allow for an increase in farmers' e-commerce participation and strengthen their interest linkage with e-commerce enterprises. It also helps to link small farmers to big markets, promotes agricultural supply-side reform, and boosts rural revitalization.

This study can be extended in several directions. This study presents an evolutionary game analysis of agricultural enterprise cooperation under government intervention, assuming the impact of government policies without practical quantification. Therefore, it would be interesting to study the impact of government-specific policy data on farmers' evolutionary stabilization strategies. In this study, government policy was considered to be deterministic. However, in other cases, government policy may be uncertain. Therefore, the introduction of a three-way evolutionary game will enrich the research. In this study, through the structural equation model, we studied what factors promoted farmers' willingness and behavior and found that government policies had no direct impact on farmers' willingness to participate, which is inconsistent with the expected hypothesis. Next, we can expand the model, reverse study which factors hinder farmers' participation, and conduct a comparative analysis.

**Author Contributions:** Conceptualization, X.W.; methodology, X.W.; software, X.W.; validation, X.W.; formal analysis, X.W.; investigation, J.R.; resources, X.W. and J.R.; data curation, X.W. and J.R.; writing—original draft preparation, X.W.; writing—review and editing, X.W.; visualization, X.W.; supervision, J.R.; project administration, J.R.; funding acquisition, J.R. All authors read and agreed to the published version of the manuscript.

**Funding:** This research was funded by the National Nature Science Foundation of China (71973106, 72271202); Shaanxi Science Fund for Distinguished Young Scholars under grant 2021JC-21; Key Scientific Research Projects of Shaanxi Provincial Department of Education under grant 21JT043; SCO Institute of Modern Agricultural Development Program under grant 4, Northwest A&F University; and Tang Scholar of Northwest A&F University.

**Institutional Review Board Statement:** Not applicable.

**Data Availability Statement:** Not applicable.

**Acknowledgments:** I am very grateful for the support of Northwest A&F University.

**Conflicts of Interest:** The authors declare no conflict of interest.

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
