# Peer review of "Influences of Government Policies and Farmers’ Cognition on Farmers’ Participation Willingness and Behaviors in E-Commerce Interest Linkage Mechanisms during Farmer–Enterprise Games"

_agriculture, doi:10.3390/agriculture12101625_

Round 1

Reviewer 1 Report

This study explores the interaction between farmers and enterprises through an evolutionary game theory model considering government intervention. Numerical analyses are provided. This paper should be improved by addressing the following comments.

 1. The novelties of this study compared to the relevant literature are not explained in the Introduction section. The authors should clearly explain how this study contributes to the related literature.

 2. In addition, prior studies are not well-reviewed and it is not clear how this study is different from and similar to the literature. Moreover, recently related studies in the fields of evolutionary game theory and e-commerce are not reviewed in this study. Hence, the authors should compare their study with the following recently related studies.

- Johari, M., & Hosseini-Motlagh, S. M. (2022). Evolutionary behaviors regarding pricing and payment-convenience strategies with uncertain risk. European Journal of Operational Research, 297(2), 600-614.

- Hosseini-Motlagh, S. M., Johari, M., Nematollahi, M., & Pazari, P. (2022). Reverse supply chain management with dual channel and collection disruptions: supply chain coordination and game theory approaches. Annals of Operations Research, 1-34. https://doi.org/10.1007/s10479-022-04909-8.

- Hosseini-Motlagh, S. M., Johari, M., & Pazari, P. (2022). Coordinating pricing, warranty replacement and sales service decisions in a competitive dual-channel retailing system. Computers & Industrial Engineering, 163, 107862. https://doi.org/10.1016/j.cie.2021.107862.

 3. In Equation 2, the second term should be changed to (1-y)*(R2+S). The authors should check and revise the other equations, which are related to Equation 2.

 4. At the end of Section 5, the authors should clearly discuss managerial insights and new findings, which are achieved in their study compared to the prior studies.

 5. The authors do not discuss future research paths at the end of the Conclusion section. The authors should clearly explain future research directions.

Author Response

26 September 2022

Dear reviewer,

We would like to thank you for the helpful comments on our paper. Here are our responses to your comments. The comments are numbered and in italics. Moreover, we highlight the revisions in the manuscript using blue font.

Thanks for your attention and my best regards

Xiaolu Wei, Junhu Ruan*.

Reviewer 2 Report

Thanks to the authors for this interesting paper with an important topic. Although the paper is well-written it might need a revision to bring out the foundation and statement more. I hope my comments help to improve the paper. Good luck!

1. Introduction:

The studies mentioned refer mainly to Asian authors and thus leave out relevant literature (e.g.  10.1016/j.compag.2022.106942 or https://doi.org/10.1016/j.compag.2018.05.032). There are many more studies, please do a more comprehensive literature review.

Line 76 ff.: Please explain what the F stands for. Factor?

2. Research Hypotheses and Theoretical Models

This is also true for the second chapter. The reference for “Farmers’ cognition” does not really fit or please be more concrete about why contract farming might be important for farmers’ cognition.

Please add the number of each hypothesis for which your references are used to derive it. At the moment, it is not clear what the hypotheses in Figure 1 are based on.

3. Variable Description and Model Construction

Line 150 the authors write: “The indicator of participation behaviors had 0–1 variables” – what does it mean when there is 0 variables? Or do you mean the characteristics/ values of the variable?

Table 2: How was the “Average” of “Daily online hour” (=4.18) built? There are only variable values from 1-4?! Additionally, for some variables, it would be more interesting to show frequencies instead of means (e.g., level of education, age, Planting year). And what were the exact question texts for the variables “Cognition of EIL mechanisms” and the following?

Figure 2: Please add a description of the three figures shown – what are the different settings? That it should be understandable by the figure.

Figure 4/ Table 7: Please add the definition for F1-F6 as the reader might not remember

5. Results Analysis

The results are not discussed with existing literature. Please add to what extent your results complement or contradict existing literature and classify your results in the existing studies. This would be especially interesting for H7 à what does that mean and how can you explain this?

Author Response

(The authors gave the same response as above.)

Reviewer 3 Report

Average values of Daily online hour, Subsidies, and Training factors in Table 2 are incorrect, should be checked

What is the volume of e-commerce?

It should be stated what policies the government has on e-commerce

It should be stated which agricultural products of the sample group are subject to e-commerce.

Author Response

(The authors gave the same response as above.)

Round 2

Reviewer 1 Report

No further comments. 

Author Response

29 September 2022

Dear reviewer,

We would like to thank you for the helpful comments on our paper. Here are our responses to the comments. The comments are numbered and in italics. Moreover, we highlight the new revisions in the manuscript using red font. 

Thanks for your attention and my best regards

Xiaolu Wei, Junhu Ruan*.

Reviewer 3 Report

It should be stated what policies the government has on e-commerce

What is the volume of e-commerce in the sample considered?

It should be stated which agricultural products of the sample group are subject to e-commerce.

Author Response

(The authors gave the same response as above.)
